# Association of Early and Severe Early Childhood Caries with Oral Health-Related Quality of Life: A Cross-Sectional Survey

**DOI:** 10.3390/healthcare13233153

**Published:** 2025-12-03

**Authors:** Samaa K. Redwan, Najlaa M. Alamoudi, Osama M. Felemban, Amani A. Al Tuwirqi, Rana A. Alamoudi

**Affiliations:** 1Pediatric Dentistry Department, Faculty of Dentistry, King Abdulaziz University, Jeddah 21589, Saudi Arabia; samaa.redwan@hotmail.com (S.K.R.); nalamoudi2011@gmail.com (N.M.A.); omfelemban@kau.edu.sa (O.M.F.); aaltuwirqi@kau.edu.sa (A.A.A.T.); 2Specialist Pediatric Dentistry, Private Practice, Jeddah 23421, Saudi Arabia

**Keywords:** preschool children, prevalence, oral health related quality of life, early childhood caries, dietary habits, oral health hygiene

## Abstract

**Background/Objectives**: Early childhood caries (ECC) and severe early childhood caries (S-ECC) are oral health problems that affect many preschool children worldwide. ECC and S-ECC negatively impact the quality of life of preschool children, including functional, psychological, and social well-being, as well as their families. There is no updated data regarding the prevalence of ECC and S-ECC in Jeddah, Saudi Arabia. Additionally, no study has been conducted on oral health-related quality of life (OHRQoL) with regard to ECC and S-ECC among preschool children in Jeddah. Thus, this investigation aimed to determine the prevalence of ECC and S-ECC in children 3 to 5 years old in Jeddah and its relation to OHRQoL. **Methods**: A cross-sectional survey was carried out on 322 children randomly selected from different preschools in Jeddah. The early childhood oral health impact scale (ECOHIS) questionnaires were distributed and completed by the parents followed by clinical examination. The dmft and dmfs index scores were determined by WHO diagnostic criteria. **Results**: A total of 322 preschool children were included in the final sample. The results indicated a caries prevalence rate of 74.2% with ECC and S-ECC prevalence rates of 34.2% and 40.1%, respectively. The mean ± SD dmft and dmfs scores were 4.6 ± 0.4 and 10.8 ± 0.9, respectively. The mean ± SD total score of the early childhood oral health impact scale (ECOHIS) was 6.1 ± 7.3. S-ECC was significantly associated with higher ECOHIS scores (*p* < 0.001) after controlling for age. **Conclusions**: ECC and S-ECC are highly prevalent in Jeddah, Saudi Arabia, and negatively impact OHRQoL.

## 1. Introduction

Early childhood caries (ECC) is defined as the presence of decayed, missing, or filled teeth (≥1) prior to the age of 71 months. For children aged 3–5, decayed, missing, or filled teeth (≥1) or a decayed, missing, or filled score of >4 (3 years), >5 (4 years), or >6 (5 years) contributes to severe early childhood caries (S-ECC) [1]. The prevalence rate of ECC varies from 20.6% to 98.0% worldwide [2]. Globally almost 560 million young children are affected by dental caries in their primary teeth, making it the 12th most common illness among children [3]. Early childhood caries (ECC) has been found to be prevalent in Middle Eastern countries [4,5]. According to current data, dental caries prevalence in Saudi Arabia remains high at 84%. This highlights the critical need for early detection and prevention of ECC in the country’s population [6]. In Al-Jouf, Saudi Arabia, preschoolers participated in a cross-sectional study. The results showed that boys had a dmfs index of 10.25, suggesting that the ECC pattern resembled the usual pattern of nursing caries [7].

Despite being the most prevalent chronic illness in children, caries still remains one of the most untreated diseases in many nations [8]. In addition to augmenting the risk of developing novel carious lesions during adulthood, untreated S-ECC and ECC result in discomfort, missed school, poor dietary behavior, and low self-esteem. All of these repercussions have a negative impact on the health and oral health-related quality of life (OHRQoL) of children [9]. Likewise, children who have early childhood caries may also experience related issues, for example, impaired development, psychological issues, and mouth discomfort that can cause trouble eating and sleeping; local infections; and a higher chance of caries in permanent dentition [10]. Other health issues linked to ECC include localized discomfort, infections, and abscesses. This can cause difficulties in chewing, malnourishment, and sleeping [11].

Globally ECC and S-ECC continue to be a source of social and economic liability [8]. They have also been demonstrated to have an impact upon a child’s quality of life [12]. For this reason, anybody who is concerned about enhancing public health must focus their efforts on preventing these issues. A parental proxy tool called the early childhood oral health impact scale (ECOHIS) is used to evaluate how early dental treatment experiences and oral health conditions affect young children’s quality of life [13]. As far as we are aware, there is no current information available on the prevalence of ECC and S-ECC and oral health-related quality of life (OHRQoL) in preschoolers in Jeddah. Considering all these factors, an effort has been made in this study to ascertain the prevalence of ECC and S-ECC in children aged 3 to 5 in Jeddah and its association with OHRQoL.

## 2. Materials and Methods

This cross-sectional study included preschool children in Jeddah, Saudi Arabia, and was conducted from November 2019 to January 2022. Data collection was paused for 7 months from March 2020 until October 2020 because of the COVID-19 pandemic. Due to having virtual classes in most public preschools, the data collection contribution was higher from private preschools. Prior to the commencement of the study, ethical approval was granted by the Research Ethics Committee at the Faculty of Dentistry, King Abdulaziz University, Jeddah, Saudi Arabia (Approval No. 096-06-19 on 18 September 2019). Moreover, the Ministry of Education also gave approval for this study.

Assuming a total of 50,000 preschool children in Jeddah and a dental caries prevalence of 73% [14], the desired sample size calculated was 323, with 80% confidence, a design effect of 2.5 to account for clustering, and 5% confidence limit. Children from private and public preschools in Jeddah were randomly selected based on four distinct geographic regions (east, west, south, and north districts). A multistage stratified random sampling technique was employed in this study to predict fair estimation of dental caries prevalence from a demographically and geographically diverse sample with minimum sample random error. The information about public and private preschools was provided by the Saudi Ministry of Education and Training. A total of eight strata were listed alphabetically in view of the public and private preschools in each district (north, south, east, and west). From each stratum, 2 preschools were randomly selected (a total of 16) to be visited for data collection. All children in the selected schools were eligible to be included in the study. The randomly selected preschools were visited twice, the first visit for the distribution of the questionnaires and the second visit for oral examination (for children whose parents had agreed and answered the questionnaire). For the inclusion criteria, healthy children aged 3 to 5 years whose parents had given consent were selected. Children with co-morbidities as well as those whose parents had refused to submit an informed consent form were excluded from the study. A total of 625 questionnaires were distributed. The study participants (parents) were directed to submit an informed consent (Arabic version) prior to the study. The parents were also provided with an explanation of the study objectives. The parents were asked to fill in and return an Arabic questionnaire. Of the 625 questionnaires, 391 (62.6%) were returned. The questionnaire comprised multiple-choice questions in two parts. The first part was about sociodemographic information such as age, gender, and the level of education of the parents. The second part contained oral health-related quality of life questions from the validated Arabic version of the early childhood oral health impact scale (A-ECOHIS) questionnaire that was validated in Arabic by Farsi et al. [15]. This part of the questionnaire comprised 13 questions divided into 2 sections: child impact and family impact. The response options were coded as follows: 0, 1, 2, 3, 4, and 5 which corresponded to never, hardly ever, occasionally, often, very often, and don’t know, respectively. A total ECOHIS score was calculated for each subject by summing the question responses except for “don’t know” responses, which were considered missing and were not added to the total ECOHIS score. The possible range for the total score was from 0 to 52. The study participants were asked to answer questions based on the whole life span of the child.

Decayed, missing, and filled primary teeth and surfaces were documented using the index by the WHO (2013) to investigate the prevalence of ECC and S-ECC. The intraoral examination was carried out according to WHO criteria in a well-lit natural light environment using disposable sterile dental mirrors and an explorer. Of the 391 children whose parents answered the questionnaire, 322 (82.4%) of them agreed and cooperated with the oral examination. The examinations were conducted by one trained examiner. The examinations were performed during the break time in the classroom while participants were seated on a chair with a back rest. No X-rays were taken. A bag that contained a toothbrush, toothpaste, oral hygiene and diet instructions pamphlet, and coloring book was distributed to each participant after the examination had concluded.

An intra-examiner reliability test was performed for dental caries diagnosis. To test intra-examiner reliability for caries, 10 pediatric dental patients were examined to measure dental caries using WHO criteria. The same patients were re-examined two weeks later, and the results of dmft and dmfs recorded by the examiner were determined using the Intraclass correlation coefficient (ICC) and found to be 0.987 (*p* < 0.001), indicating excellent reliability.

### Statistical Analysis

SPSS software (SPSS, Inc., Chicago, IL, USA) version 20 was used for statistical analysis, taking *p*-values < 0.05 as statistically significant. A chi-square test was performed for the categorical variable to assess whether significant associations were observed between groups. The Shapiro–Wilk test in addition to visual inspection of Q-Q plots were employed to test the data normality, and the data was found to be non-normally distributed. The chi-square test, Independent-Samples Mann–Whitney U test, and Kruskal–Wallis test were used to compare data between groups. Finally, an adjusted multiple linear regression analysis was modeled to control for possible confounding in the association between caries diagnosis (independent variable) and ECOHIS (dependent variable).

## 3. Results

Of the 625 families that were contacted, 322 completed the questionnaire, and their children were clinically examined (response rate of 51.5%). The mean ± SD age of the subjects was 4.6 ± 0.8 years, and more than half of them were males (58.1%). The number of participants in public and private preschools was 64 (19.9%) and 258 (80.1%), respectively. The parental education level data exhibited that 79.2% and 77% of respondents had reached the university/college educational level, respectively.

The association between demographics and dental caries diagnosis is shown in Table 1. More than two thirds of the 3–4-year-olds (38.8%) were caries-free compared to only 15.8% of the 5-year-olds. On the contrary, 18.7% of the 3–4-year-olds had ECC compared to 45.9% of the 5-year-olds. S-ECC accounted for similar percentages in the 3–4-year-olds (42.4%) and 5-year-olds (39.3%). Table 1 also shows the distribution of the early childhood oral health impact scale (ECOHIS) scores by age, gender, school type, and parents’ education. There was a significant association between age and total ECOHIS score (*p* = 0.004). Older subjects (5 years old) had significantly higher mean scores than younger subjects (3–4 years old). In regard to gender, there was no significant difference in impact between males and females (*p* = 0.796). Moreover, preschool type had a significant association with total ECOHIS score (*p* = 0.001). Subjects from public preschools had higher ECOHIS scores (9.7 ± 8.6) than subjects from primary preschools (6.2 ± 6.7). The mean ± SD dmft and dmfs of the sample were 4.6 ± 0.4 and 10.8± 0.9, respectively.

The association between caries and ECOHIS is shown Figure 1. There was a significant association between the dental caries diagnosis and ECOHIS (*p* < 0.001). ECOHIS scores increased as the dental caries diagnosis increased in severity. A post hoc analysis revealed that the ECOHIS score of the S-ECC group (8.8 ± 7.6) was statistically significantly higher than that of the ECC group (6.4 ± 7.5) and the caries-free group (4.6 ± 5.5).

A multiple linear regression analysis was modeled to evaluate the effect of caries on quality of life (QoL) while controlling for possible confounding by age (Table 2). Children having S-ECC had significantly greater ECOHIS scores (lower QoL) on average by 3.8 compared to children with no caries controlling for age, and the increase was statistically significant (*p* < 0.001). Children with ECC had higher ECOHIS scores by 0.8 controlling for age, but the result was not statistically significant (*p* = 0.437). Children aged 5 years had higher ECOHIS scores (lower QoL) by 2.4 compared to children aged 3 to 4 years controlling for caries, and the increase in ECOHIS score was statistically significant (*p* = 0.004).

## 4. Discussion

ECC and S-ECC are regarded as morbid diseases that are challenging to treat in infants and young children. This study examined the frequency and severity of caries in preschoolers. It also assessed how oral health-related quality of life (OHRQoL) is connected to dental health in Jeddah, Saudi Arabia. In this study, 16 randomly chosen preschools run by public and private managements in four distinct Jeddah districts were selected. A total of 322 children were examined to find out the prevalence of ECC and S-ECC. The study showed that the overall prevalence of caries was 74.2%. The prevalence of ECC was 34.2%, and that of S-ECC was 40.1%. The high dental caries prevalence among preschoolers in our sample was in line with other research works conducted in Saudi Arabia [16,17,18] and the UAE [19]. However, the prevalence rate of our study was in contrast to that observed by Abdellatif et al. and Alhabdan et al. in Riyadh. It was equivalent to 83%, while the highest in Saudi Arabia was reported by Al-Rafee et al. (85.77%) [20,21,22]. A cross-sectional survey carried out among primary male school children in Riyadh, Saudi Arabia, determined the prevalence to be 80.15% [23]. Moreover, the current study’s observed prevalence of dental caries in children was much higher than the WHO/FDI objective of 50% [24]. In our investigation, mean dmft and dmfs scores were 4.6 ± 0.4 and 10.8 ± 0.9, respectively. Similar values of dmft were shown in earlier studies carried out in Dawadmi, Saudi Arabia (3.69), and Qatar (3.3) [4,25]. The dmft value in the current study was slightly greater than those reported by earlier studies conducted in Al-Madinah, Saudi Arabia (2.66) [26]. The disparity between the mean dmft and the prevalence of caries might be caused by varying degrees of preventative care undertaken in various regions as well as geographical variations in the environment, society, and culture. The combined data from our investigation and other research indicates that dental caries is widespread in the Middle Eastern population and poses a threat to public health. This emphasizes the fact that dental caries is still a major problem among preschoolers in Saudi Arabia.

ECOHIS is a validated tool designed to be used in epidemiological surveys to explore the effects of oral health as well as dental diseases on preschoolers’ and their families’ quality of life [15]. The ECOHIS scores and the occurrence of dental disease correlated as predicted; that is, the more severe the dental disease, the higher the ECOHIS scores. A substantial association was discovered between the caries experience and the ECOHIS scores. The null hypothesis was rejected in the current investigation since there was a difference in mean ECOHIS scores between children with S-ECC, children with ECC, and children without caries. Children with S-ECC showed a considerably higher ECOHIS scores than children with ECC, which was followed by subjects without caries. Children with more caries experiences had increased odds of experiencing unfavorable effects in relation to the relationship between OHRQoL and dental caries. Research from several geographical locations, including Brazil [27,28,29], India [30], and the United Kingdom [31,32], supports this finding. This outcome was comparable to a study conducted in East China, which found that children with S-ECC had considerably greater ECOHIS scores than children without S-ECC [10]. Children in public preschools scored significantly higher on the ECOHIS compared to those in private preschools, suggesting they had a poorer oral health-related quality of life. Disparities in oral health may be influenced by underlying socioeconomic factors, dental care accessibility, and parents’ oral health habits. The smaller number of public preschool children in our sample, especially following COVID-19 restrictions, calls for careful interpretation and underscores the necessity for more comprehensive, stratified research to validate these results.

Dental caries is linked with numerous risk factors in young children, such as the time/date of the child’s first dental visit, prolonged or at-will breastfeeding, prolonged/frequent/nocturnal bottle feeding, oral hygiene practices, and dietary habits [21]. Behavioral factors, such as eating a diet heavy in sugar and poor dental hygiene, contribute as risk factors [33]. Furthermore, most Saudis only go to the dentist when they are in discomfort rather than on a regular basis for examinations [34]. In a recent survey carried out in Jeddah, it was shown that 1 in 4 children have never visited a dentist [35]. Additionally, research has looked at the amount of fluoride in drinking water supplies in Saudi Arabian regions and discovered that it is lower than recommended [36]. Preschool children lack the mental capacity and manual dexterity necessary to practice proper dental hygiene. In this regard, parental support and supervision are crucial to lowering the chance of developing caries. The practice of toothbrushing should be performed by a parent twice a day using an appropriately sized toothbrush. In the current study, apart from age, which revealed a statistically significant association with dental caries, the demographic parameters did not show any association. The prevalence of caries increased as age increased, which is consistent with other investigations [37]. This conclusion may be attributed to the children’s nutrition and dental hygiene practices, which change as they become older, increasing the cariogenic risk [38].

This study’s strength lies in the fact that it is the first to directly compare caries-free children with those with ECC and S-ECC. This cross-sectional study in Jeddah, Saudi Arabia, discovered a positive association between dental caries and a deterioration in children’s oral health-related quality of life. However, a well-designed longitudinal study could provide valuable information and obtain deeper insight. The study was carried out in a single city; thus, the results cannot be applied to the entirety of Saudi Arabia, limiting its generalizability. Additionally, the sample may not have been as representative as it might have been because only a small number of the participants from public preschools were investigated, and only private preschools were visited after the COVID-19 lockdown. Furthermore, we are unable to directly link the outcomes to a cause-and-effect relationship. Additionally, the use of proxy reports limits the findings for preschoolers’ OHRQoL. The severity of oral disease in children’s dental health may be understated in proxy reports. In addition, only variables that were significantly associated with dental caries (exposure) and also with the ECOHIS (outcome) were included in the regression model, which may have led to the exclusion of other confounders and may have resulted in residual confounding. Also, using linear regression for an outcome that is not normally distributed is a limitation, and alternative models may have offered a more suitable fit for the data. However, because this was a preliminary investigation, the authors recommend that more studies be conducted with a greater sample size to provide solid evidence for the current study’s findings regarding the OHRQoL of preschool-aged children before and after dental treatment.

## 5. Conclusions

This research has expanded the body of knowledge about the impact of ECC, particularly in its severe form, on a child’s overall health and well-being, with a higher prevalence rate of 74.2%, though the findings may have underestimated the true picture of ECC and S-ECC in the population. S-ECC children had notably higher ECOHIS scores (lower OHRQoL). Our study raises the alarm about the pressing need for our society to adopt preventative and therapeutic dental health programs for young children. Preschoolers who participate in these programs should have better oral hygiene habits and eat less sugary snacks.

## Figures and Tables

**Figure 1 healthcare-13-03153-f001:**
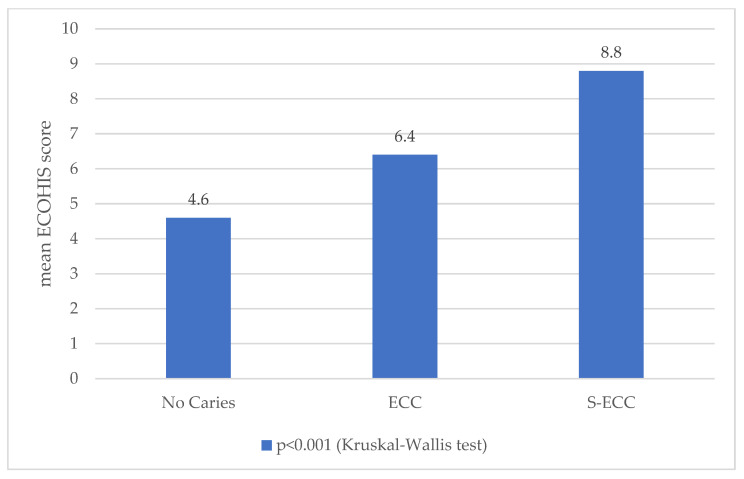
The association between dental caries diagnosis and the oral health-related quality of life (ECOHIS).

**Table 1 healthcare-13-03153-t001:** Association between demographic characteristics and caries diagnosis and oral health-related quality of life.

Demographics		n	Caries Diagnosis				Quality of Life	
			No Caries	ECC	S-ECC	*p*-Value €	ECOHIS	*p*-Value †
			n (%)				Mean ± SD	
Age	3–4 years	139	54 (38.8)	26 (18.7)	59 (42.4)	<0.001 *	5.5 ± 6.0	0.004 *
	5 years	183	29 (15.8)	84 (45.9)	70 (39.3)		8.0 ± 8.0	
Gender	Male	187	52 (27.8)	58 (31.0)	77 (41.2)	0.342	7.0 ± 7.0	0.796
	Female	135	31 (23.0)	52 (38.5)	52 (38.5)		6.9 ± 7.6	
Preschool Type	Public	64	16 (25.0)	28 (43.8)	20 (31.3)	0.156	9.7 ± 8.6	0.001 *
	Private	258	67 (26.0)	82 (31.8)	109 (42.2)		6.2 ± 6.7	
Mother’s Education	High school or less	67	10 (14.9)	27 (40.3)	30 (44.8)	0.072	7.6 ± 8.1	0.682
	University/college	255	73 (28.6)	83 (32.5)	99 (38.8)		6.7 ± 7.0	
Father’s Education	High school or less	74	17 (23.0)	23 (31.1)	34 (45.9)	0.499	6.8 ± 7.3	0.920
	University/college	248	66 (26.6)	87 (35.1)	95 (38.3)		7.0 ± 7.3	

ECC = early childhood caries. S-ECC = severe early childhood caries. ECOHIS = early childhood oral health impact scale. n = number of subjects. SD = standard deviation. € = chi-square test. † = Independent-Samples Mann–Whitney U test. * = statistically significant (*p* < 0.05).

**Table 2 healthcare-13-03153-t002:** Multiple linear regression analysis model with ECOHIS questionnaire score (dependent variable).

Independent Variables		β ± SE	95% CI	*p*-Value
Age	5 years	2.4 ± 0.8	0.8–4.1	0.004 *
	3–4 years	Reference		
Caries	S-ECC	3.8 ± 1.0	1.8–5.8	<0.001 *
	ECC	0.8 ± 1.1	−1.3–2.9	0.437
	No caries	Reference		

SE: standard error. CI: confidence interval. * significance level < 0.05.

## Data Availability

The raw data supporting the conclusions of this article will be made available by the authors on request.

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
