# Peer review of "Association of Early and Severe Early Childhood Caries with Oral Health-Related Quality of Life: A Cross-Sectional Survey"

_healthcare, 2025, doi:10.3390/healthcare13233153_

Round 1
Reviewer 1 Report
Comments and Suggestions for Authors
Dear authors,
This manuscript investigates the prevalence of early childhood caries (ECC) and severe ECC (S-ECC) among 3–5-year-old children in Jeddah and examines their impact on oral health-related quality of life using the Arabic version of ECOHIS. The topic is relevant for pediatric oral health and public health, and the overall message, that S-ECC is associated with worse OHRQoL than ECC and caries-free status, is clear and supported by the data.
However, several methodological clarifications, consistency issues, and some aspects of the statistical analysis and discussion need to be addressed before the manuscript is suitable for publication. I therefore recommend major revision.
Major comments
- Sample size calculation and precision indicators
The sample size calculation is described as being based on a prevalence of 73% with “80% confidence and 5% confidence limit”. This is unusual, since cross-sectional epidemiological studies typically use a 95% confidence level.
- Please clarify whether “80%” is correct and, if so, justify this choice. If it is a typographical error, it should be corrected throughout.
- It would also be helpful to state whether a design effect was considered given the multistage stratified sampling.
- Finally, please explain how you arrived at the final number of 322 participants (e.g., initial approached sample, refusals, exclusions). A brief flow description (even if only in text) would improve transparency.
- Representativeness and selection bias
The final sample is heavily skewed towards private preschools, partly due to the COVID-19-related interruption of data collection. This limitation has important implications for the generalizability of prevalence estimates and potentially for the association with OHRQoL (as school type is a proxy for socioeconomic status).
- Please describe more explicitly how many children were ultimately recruited from public vs. private preschools and how this differs from your original sampling plan.
- In the Discussion, strengthen the limitations section to explain how this imbalance might bias the prevalence estimates and the observed relationships with OHRQoL.
- If feasible, you could briefly comment on differences in caries experience and ECOHIS scores between public and private schools (you already partially present this in the tables) and interpret them in light of socioeconomic differences.
- Handling of “don’t know” responses in ECOHIS
The description of the ECOHIS response options includes “don’t know”, but the manuscript does not explain how these responses were handled in scoring. In many ECOHIS studies, “don’t know” is treated as missing and excluded from the total score.
- Please specify clearly how “don’t know” answers were processed: were they coded as a specific value or treated as missing?
- If they were included numerically, this should be justified and the potential impact on total scores discussed. If they were treated as missing, please mention any rules used (e.g., allowable number of missing items per scale).
- Inconsistencies in reporting dmft/dmfs values
There appears to be an inconsistency between the dmft/dmfs values reported in the Abstract and those in the Results section. For example, dmft is reported with a very small standard deviation in one place and a much larger standard deviation in another.
- Please check the dmft and dmfs mean and variability values and make sure they are consistently reported across Abstract, Results, tables, and figures.
- Also specify explicitly whether the values presented are means with standard deviations or means with standard errors.
- Choice and specification of the multivariate model
The regression model assessing the association between caries status (no caries / ECC / S-ECC) and ECOHIS score appears to include adjustment only for age. This is very minimal and may not be sufficient to control for confounding. School type and parental education, at a minimum, are likely to be important covariates.
- Please reconsider the multivariate model and, if sample size permits, adjust for additional relevant variables (e.g., age, gender, school type, parental education).
- If you decide not to include more covariates, provide a clear rationale and discuss this as a limitation (residual confounding).
- Model choice for ECOHIS scores
ECOHIS is a summed score derived from ordinal items and, as you correctly report, shows a non-normal distribution. While regression with a continuous outcome is common in practice, alternative models (e.g., Poisson or negative binomial regression) may be more appropriate for count-like outcomes.
- At minimum, please acknowledge in the Discussion that the use of linear regression for a non-normally distributed, count-like outcome is a limitation and that alternative models might provide a better fit.
- If you have access to the raw data and it is feasible, you might consider confirming the main findings using a Poisson/negative binomial or ordinal regression model and reporting whether the conclusions remain unchanged.
- Focus and structure of the Discussion
The Discussion section is relatively long and sometimes reiterates general background information that has already been presented in the Introduction. The main findings and their implications could stand out more clearly.
- I suggest condensing the background portions and focusing more on:
- how your prevalence and OHRQoL findings compare quantitatively with previous studies in Saudi Arabia and the region,
- the clinical/functional meaning of the observed difference in ECOHIS scores (e.g., how a certain increase translates to everyday impact for the child and family),
- concrete implications for preventive strategies, screening, and policy.
Minor comments
- Terminology and wording
- Terms such as “KG schools”, “primary preschools” and similar expressions should be checked for consistency and clarity. Consider using “public preschool” / “private preschool” or “public kindergarten” / “private kindergarten” consistently throughout the manuscript to avoid confusion with “primary school” (elementary).
- Please check for occasional typographical errors (e.g., dmfs vs. “smfs”) and correct them.
- Language and style
- Some sentences in the Introduction and Discussion are quite long and could be broken up to improve readability. A light language edit focusing on sentence length, punctuation, and avoidance of repetition would help.
- Avoid repeating the same general statements about ECC prevalence and severity multiple times; one clear, well-referenced statement in the Introduction is sufficient.
- Abstract
- Indicate explicitly the final number of children included in the analysis and, if space allows, mention briefly the main limitation related to school type distribution (public vs. private).
- Ensure the dmft/dmfs figures in the Abstract match exactly those in the Results once corrected.
- Reporting of statistical tests
- In figures and tables, make sure each reported p-value is clearly associated with the test used (e.g., “Kruskal–Wallis test”, “Mann–Whitney U test”). In most places this is already done; please just verify consistency.
- Limitations section
- You already mention several limitations. It might improve clarity to group them more systematically at the end of the Discussion:
- cross-sectional design,
- single-city sample with predominance of private preschools,
- reliance on parent-reported OHRQoL (proxy reporting),
- possible impact of the COVID-19 period on dental service use and oral health behaviors,
- limitations related to the model choice and covariates.
- Tables and figures
- Check that all abbreviations used in tables and figures (ECC, S-ECC, dmft, dmfs, ECOHIS, etc.) are defined either in the table/figure footnote or in the legend.
- Make sure the number of participants analyzed in each table/figure is clear, especially if any missing data occurred for ECOHIS items.
Reviewer 2 Report
Comments and Suggestions for Authors
I want to thank the authors for the good and interesting study. I have some minor comments in the attached file. Most of the comments are related to statistical analysis and presentations, but they don't affect the overall results and conclusions.

Round 2
Reviewer 1 Report
Comments and Suggestions for Authors
Congratulations, the article looks good now. I propose publishing it.